# Selected Aspects of Nutrition in the Prevention and Treatment of Inflammatory Bowel Disease

**DOI:** 10.3390/nu14234965

**Published:** 2022-11-23

**Authors:** Paulina Panufnik, Martyna Więcek, Magdalena Kaniewska, Konrad Lewandowski, Paulina Szwarc, Grażyna Rydzewska

**Affiliations:** 1Clinical Department of Internal Medicine and Gastroenterology with Inflammatory Bowel Disease Subunit, Central Clinical Hospital of Ministry of the Interior and Administration in Warsaw, 02-507 Warszawa, Poland; 2Collegium Medicum, Jan Kochanowski University, 25-317 Kielce, Poland

**Keywords:** inflammatory bowel disease, prevention, Crohn’s disease exclusion diet, ulcerative colitis exclusion diet, exclusive enteral nutrition, low FODMAP diet

## Abstract

Inflammatory bowel disease has become a global health problem at the turn of the 21st century. The pathogenesis of this disorder has not been fully explained. In addition to non-modifiable genetic factors, a number of modifiable factors such as diet or gut microbiota have been identified. In this paper, the authors focus on the role of nutrition in the prevention of inflammatory bowel disease as well as on the available options to induce disease remission by means of dietary interventions such as exclusive and partial enteral nutrition in Crohn’s disease, the efficacy of which is reported to be comparable to that of steroid therapy. Diet is also important in patients with inflammatory bowel disease in the remission stage, during which some patients report irritable bowel disease-like symptoms. In these patients, the effectiveness of diets restricting the intake of oligo-, di-, monosaccharides, and polyols is reported.

## 1. Introduction

As shown by numerous analyses, the prevalence of inflammatory bowel disease increases year by year [1,2,3]. Crohn’s disease and ulcerative colitis are increasingly diagnosed in both the pediatric and adult populations in developed as well as developing countries. At the turn of the 21st century, inflammatory bowel disease has become a global health problem. Its prevalence is on the increase in developing countries while having already exceeded the level of 0.3% in developed countries [3]. The exponential increase in the incidence of IBD worldwide indicates that environmental factors that have significantly changed over the past decades play a key role in the development of inflammatory bowel diseases [4]. The pathogenesis of either Crohn’s disease or ulcerative colitis has not been fully explained [5]. In addition to non-modifiable genetic factors, a number of modifiable factors such as gut microbiota, diet, or lifestyle have been identified. Diet shapes the gut microbiome, and it uses food ingredients for growth. The body’s cells use microbiome metabolites as immunomodulatory agents and as energy sources [6].

In this article, we focus mainly on the influence of diet on the development and course of IBD.

## 2. Gut Microbiota

The gut microbiome plays an important role in the pathogenesis of IBD. Germ-free (GF) mice have decreased numbers of regulatory T cells and T helper 17 cells in the colonic lamina propria [7]. Also evident in GF mice is a defective colonic mucus barrier, which is a barrier against luminal bacteria [8]. Colonization of the gut of mice with the gut microbiome restores mucus barrier functions and mucosal immunity.

On the other hand, there is evidence of a pro-inflammatory effect of the gut microbiome. Most GF mice do not develop colitis, like mice whose intestines have been colonized by a microbiome taken from a healthy person. By contrast, colonization of the gut with the microbiome from IBD patients causes severe colitis in mice [9,10]. This clearly demonstrates the influence of the gut microbiome on the development of IBD.

Diet plays a significant role in shaping the homeostasis of the gut microbiome. But inflammation in the gut can cause intestinal dysbiosis and adversely affect the use of food components by the microbiome and the host cells [11].

In recent years, changes within the gut microbiota of patients with inflammatory bowel disease have been demonstrated in numerous studies [12,13,14]. The composition and function of the gut microbiota are sensitive to changes in the diet and the environment that surrounds us. A general decline in microbial diversity has been observed in IBD patients. It is estimated that there are trillions of microbial cells dwelling within the human intestinal lumen, with 99% of the overall composition in healthy individuals being comprised of Bacteroidetes, Firmicutes, Proteobacteria, and Actinobacteria. The Firmicutes and Bacteroidetes together account for about 90% of the microbiome; along with the plant fiber oligosaccharide-degrading bacteria, they are responsible for the production of short-chain fatty acids (SCFAs). SCFAs are the source of energy for the colonic epithelium and play a key role in the regulation of intestinal homeostasis [15,16].

Several food-derived ingredients (e.g., phytochemicals and food additives) have been identified, especially in animal models, that affect the gut microbiome, mucosal immunity, and mucosal barriers. Through this influence, these ingredients can contribute to or prevent inflammation in the gut [11].

The influence of high consumption of simple sugars on the development of colitis has been proven in an animal model [17]. However, this effect was not observed among germ-free mice, which suggests a large influence of changes in the gut microbiome in this phenomenon. It has also been proven that the supplementation of the diet of mice with certain amino acids (e.g., arginine, tryptophan, histidine, glutamine, etc.) relieved inflammation of the colon in animals [18,19,20,21].

Animal studies cannot be extrapolated to the clinical setting. Some dietary interventions that affect the gut microbiome, such as the CDED diet (discussed later in this article), are effective both in an animal model and in humans [22]. However, for example, although effective in relieving the symptoms of CD, in animal studies, supplementation with fructo-oligosaccharides was not effective in humans [23].

Some role in the pathogenesis of IBD is also ascribed to fungi dwelling within the gastrointestinal tract. Patients with IBD were shown to present with higher numbers of *Candida albicans* and lower numbers of *Saccharomyces cerevisiae* [24]. In addition, *Malesezia limiteda* was detected in patients with Crohn’s disease. These fungi were shown to be responsible for the development of colitis in animal models [25]. However, the impact of fungi on the development of IBD and the influence of the diet on the composition of gut microbiota has been poorly examined.

The topics related to gut microbiota, its impact on the incidence of IBD, as well as the impact of diet, lifestyle, infections, or even child delivery methods on microbial composition are still under study. The impact of SCFAs, particularly butyrate, on the intensification of the inflammatory process in the course of IBD has been quite extensively described in the literature. Patients with IBD present with reduced numbers of bacteria responsible for the production of SCFAs, in particular *F. prausnitzii*, which consequently leads to a reduction in fecal SCFA levels. However, it is not clear whether this is a consequence or a cause of increased inflammation in the course of inflammatory bowel diseases [26,27].

The effect of SCFAs on the regulation of biological functions in the gut is fairly well understood. SCFAs regulate gene expression epigenetically by inhibiting HDACs (histone deacetylases) [28] and work as signaling molecules by way of GPRs (G-protein-coupled receptors) in host cells [29].

Facchin et al. [30] demonstrated the beneficial effect of sodium butyrate supplementation on the quality of life of patients with ulcerative colitis. The growth of bacteria capable of producing SCFAs with potential anti-inflammatory activity was also observed in patients with IBD after two-month supplementation with microencapsulated sodium butyrate. However, a prospective multicenter randomized placebo-controlled study carried out by Pietrzak et al. [31] in a pediatric population failed to demonstrate the efficacy of 12 weeks of supplementation with sodium butyrate as a supportive treatment in IBD patients.

Increasingly broader knowledge about the gut microbiome and its impact on the development and course of IBD prompts research on interventions in its composition. Gut microbiota targeting therapies—probiotics, prebiotics, and FMT (fecal microbiota transplantation) have been studied. However, none of these therapies is currently used routinely in the treatment of IBD [32,33]. A systematic review of randomized controlled trials to examine the efficacy of probiotics, prebiotics, and synbiotics in the treatment of IBD reveals that the use of probiotics can induce anti-inflammatory reactions and induce remission in IBD [34]. It seems that more research is needed into an effective treatment for dysbiosis that could be used in IBD.

Issues related to the gut microbiota, probiotic therapy, and SCFA supplementation are very broad scientific topics subject to ongoing research and need to be discussed separately. We mention the effects of particular dietary interventions on the composition and function of the gut microbiome later in this article.

## 3. The Role of Diet in the Etiology of IBD

Diet is an easily modifiable factor affecting the prevalence of inflammatory bowel disease. The literature contains considerable evidence on the risk of IBD being reduced by breastfeeding, and reduced intake of animal fats and proteins, as well as by following the Mediterranean diet.

The impact of breastfeeding on the reduced incidence of IBD has been proven in a number of studies [35,36,37,38]. A meta-analysis of 35 studies showed that any duration of the breastfeeding period was associated with a decrease in this risk while the greatest decrease was observed in infants breastfed for more than one year [35]. In addition, breastfeeding, just like the delivery route and the maternal diet during pregnancy, has a significant effect on the neonatal microbiome [37]. Exclusive breastfeeding for the first six months of a child’s life with a subsequent continuation of breastfeeding accompanied by breastfeeding supplementation is recommended by the European Society for Pediatric Gastroenterology, Hepatology, and Nutrition (ESPGHAN) [39].

The Western diet, which is rich in animal proteins, saturated fats, and processed foods, is associated with an increase in the risk of inflammatory bowel disease, while the Mediterranean diet, which is a diet rich in fish, nuts, and fiber is associated with a decrease in this risk [21,40,41,42,43].

The literature contains no evidence of the impact of protein consumption on the risk of IBD development. Most of the studies were retrospective in character, but the impact was also demonstrated in a prospective study carried out in a group of more than 67,000 female patients. A total of 77 patients were diagnosed with IBD over a 10-year observation period in the study. After the analysis of the nutritional habits as declared by the subjects prior to the diagnosis, the affected subjects were found to be characterized by high consumption of total protein, particularly animal protein. Interestingly, the increased risk of IBD was linked to the high consumption of animal protein from meat and fish. No correlation of this type was observed for egg and milk proteins [44]. The impact of high consumption of meat, including red meat, on the increased risk of UC was also demonstrated by Peters et al. in 2022 [45] as well as in a meta-analysis of nine studies comparing meat-eating subjects to vegetarian subjects published in 2015 [46].

A correlation between high consumption of meat and increased risk of ulcerative colitis was also demonstrated in a systematic review of the literature published in 2011 [40] and encompassing a total of 2609 patients with inflammatory bowel diseases and 4000 healthy individuals. As also shown in the same paper, high overall fat consumption is associated with an increased risk of Crohn’s disease.

The prevalence of inflammatory bowel diseases may also be associated with the consumption of polyunsaturated fatty acids (PUFAs). The *n*-3 PUFAs (docosahexaenoic acid (DHA), eicosapentaenoic acid, and alpha-linolenic acid (ALA)), present, among others, in fish and certain cold-pressed oils, were shown to have anti-inflammatory properties. In contrast, the *n*-6 PUFAs (linoleic acid (LA), arachidonic acid) present in nuts, avocados, or olive oil, were shown to have pro-inflammatory properties. Maintaining intestinal homeostasis requires that *n*-3 and *n*-6 PUFAs are consumed in an appropriate ratio. The Western diet is characterized by this ratio being shifted towards higher consumption of *n*-6 PUFAs which has been linked to increased risk of IBD [40,47,48,49]. For example, a prospective cohort study carried out to assess the effect of multiple factors, including diet, on the incidence of certain cancers and chronic diseases (European Prospective Investigation into Cancer and Nutrition, EPIC) confirmed the increased risk of ulcerative colitis in subjects with high linoleic acid intake [50].

The impact of dietary fats on the development of Crohn’s disease has also been demonstrated in numerous studies; for example, as shown by Amre et al. in 2007, consumption of *n*-3 PUFAs and more favorable *n*-3 PUFA to *n*-6 PUFA ratios were associated with a lower risk of CD in the pediatric population [51]. The protective effect of *n*-3 PUFAs, in particular DHA, has also been demonstrated in the adult population in prospective observational trials by John et al. [52] and de Silva et al. [53]. Fish oil supplementation was also demonstrated to have a beneficial effect on the risk of UC. In addition, patients receiving this supplementation presented with lower levels of C-reactive protein (CRP) and higher levels of albumin at diagnosis [54]. By contrast, a meta-analysis by Wang et al. [55] failed to confirm the relationship between total fat consumption and the risk of UC, although a decrease in morbidity was observed at higher DHA intake.

The effects of dietary fiber on the risk of inflammatory bowel disease have also been extensively described in the literature. Dietary fiber is an important component of the diet. It consists of two fractions: the soluble fraction and the insoluble fraction. As mentioned before, the soluble fraction is converted to SCFAs by microorganisms within the large bowel lumen [56]. An animal model was used to demonstrate that excessively low intake of dietary fiber results in mucus glycoproteins being used as the source of nutrition by the gut microbiota, leading to the erosion of the mucous barrier and consequently colitis [57].

The Nurses’ Health Study, which involved a 26-year observational period and was carried out in more than 170,000 female subjects, showed that higher dietary fiber intake was associated with a lower risk of Crohn’s disease and had no effect on the risk of ulcerative colitis. The decrease in disease incidence was related to the consumption of vegetable fiber whereas the fiber obtained from cereals and leguminous plants had no effect on the disease risk [58]. No effect of dietary fiber on the risk of inflammatory bowel disease was demonstrated in the EPIC IBD study [59].

Dietary fiber may affect the length of remission in patients with ulcerative colitis. The effect of supplementation with dietary fiber from ribwort plantain (*Plantago lanceolata*) on the maintenance of CU remission was similar to that of mesalazine, as demonstrated by Fernández-Bañares et al. [60]. However, limited evidence on the beneficial effect of dietary fiber on IBD was found in a systematic review of 23 randomized studies [61].

The risk of inflammatory bowel disease in subjects following Western dietary habits that include a high content of processed foods may also be increased by food additives used therein. Emulsifiers, thickeners, and sweeteners are extensively added to food products. One such substance is carrageenan used as a thickener in the manufacture of jellies, dairy products, or dressings. Carrageenan affects the structure and function of the intestinal epithelium by damaging the enterocytes’ tight junctions, promoting increased permeability of macromolecules across the intestinal mucosa and contributing to increased inflammation [62]. Other food additives with established proinflammatory properties include carboxycellulose and polysorbate-80, which are used as emulsifiers and stabilizers e.g., in the production of ice cream. These substances are responsible for the increased proinflammatory potential of the microbiome, increased bacterial adhesion to the intestinal mucosa, and increased migration of bacteria across the intestinal epithelium [63,64]. 

In addition to food additives, inflammation within the intestinal lumen can also be promoted by natural ingredients of food products. Examples of such ingredients are gluten and amylase/trypsin inhibitors (ATI), both to be found in wheat [65,66]. ATIs promote immune response by activating Toll-like receptors on myeloid cells [67] while gluten induces the release of zonulin which is linked with enterocytes’ tight junctions and increased interleukin 10 levels [68].

In addition to the previously described impact of the Western diet on the development of IBD, its impact on the development of obesity, which affects 1/3 of UC and CD patients, is also significant [69,70]. The cause of excess body weight in IBD patients may be similar to that in the rest of society—excessive food intake—or may be related to, for example, glucocorticosteroids. ESPEN and UEG have issued joint recommendations, namely the “European guideline on obesity care in patients with gastrointestinal and liver diseases” [71], which recommend that every patient with IBD and obesity should assess fatty liver, exclude insulin resistance and obesity-related diseases such as diabetes, hypertension, and dyslipidemia. These guidelines also highlight the need to reduce body weight in obese patients during disease remission to reduce the risk of perioperative complications and enhance response to therapy with biologicals. Weight reduction should be supervised by someone experienced in the nutritional treatment of people with IBD.

## 4. The Role of Enteral Nutrition in the Management of Inflammatory Bowel Diseases

Exclusive enteral nutrition (EEN) is the best-tested approach to nutritional management aimed at inducing remission in mild and moderate Crohn’s disease. EEN is recommended by the European Crohn’s and Colitis Organization (ECCO) and ESPGHAN as the treatment of choice in pediatric patients [72]. The premise of exclusive enteral nutrition consists of complete liquid formula being administered to the patient as the only source of nutrition for 6 to 8 weeks. Nutrition may be administered orally or via a nasogastric tube. A response to the dietary treatment should be observed within 2 weeks. As the intervention is completed, the patient should gradually expand their diet by including solid products for the next 2 to 3 weeks [72,73].

The efficacy of exclusive enteral nutrition in achieving clinical remissions in the pediatric population was comparable to the efficacy of glucocorticosteroids and amounted to about 80%, whereas intestinal mucosal healing was observed more frequently in patients in the EEN group [74,75,76,77]. In addition, EEN is a safer therapeutic modality. The most severe complication observed in patients receiving EEN consisted of the refeeding syndrome, observed in patients presenting with extreme undernutrition prior to the treatment. However, this complication can be counteracted by monitoring at-risk patients [78]. Other adverse effects of EEN include diarrhea, constipation, bloating, and nausea [79].

Mechanisms responsible for the effectiveness of EEN include anti-inflammatory effects, recovery of the intestinal epithelial barrier, bowel rest, and modification of the intestinal microbiome [80]. Although the alteration in the gut microbiome due to EEN is said to have a beneficial effect in alleviating inflammation in the gut, the exact mechanism behind this effect is unknown. EEN reduces the diversity of the gut microbiome and reduces the amount of Faecalibacterium and Bifidobacterium, which are bacteria desired in the gut lumen [81]. The reason for the beneficial effects of EEN on the intestinal mucosa, despite the apparently unfavorable changes in the composition of the intestinal microbiome, requires further research.

The high efficacy of exclusive enteral nutrition in terms of achieving remission of Crohn’s disease in the pediatric population encourages the use of this approach in the adult population. A literature review by Wall et al. [82] points to the comparable efficacy of EEN and glucocorticosteroids in terms of adult remissions while drawing attention to the high percentage of patients who fail to complete the treatment. Depending on the study, this could be as much as one-half of the subjects. The low rate of compliance with EEN recommendations was usually due to the flavor of the nutritional formula and patients’ unwillingness to feed exclusively on it [83]. For this reason, studies were undertaken to assess the efficacy of partial enteral feeding as regards the achievement of Crohn’s disease remission.

Partial enteral nutrition (PEN) combines liquid formula and solid foods [76]. Its efficacy was examined in 2006 by Johnson et al. [84] in a study involving 50 pediatric patients who were assigned to two study groups receiving either EEN or PEN (with 50% of the caloric demand originating from the liquid formula, and 50% from any type of oral diet as per the patient’s preference). A much higher efficacy was demonstrated for EEN as compared to PEN, with the efficacy of PEN being as low as 15%. The researchers also observed an increase in albumin and hemoglobin levels in the group receiving exclusive enteral nutrition. After the caloric demand met by the enteral formula in the partial enteral nutrition was increased to 80–90%, the efficacy of PEN was raised to 65% and the compliance rate was 87% [85]. Due to the greater acceptability of PEN as compared to EEN, the former appears to be a promising option for treatment [86]. However, partial enteral nutrition is not yet recommended as routine treatment in patients with inflammatory bowel diseases due to the insufficient amount of data available.

Due to the unsatisfactory response to partial enteral nutrition combined with a diet of preference and the difficulties with compliance with exclusive enteral nutrition, a diet has been developed that is based on generally available foods and excludes potentially proinflammatory products. The diet is referred to as the Crohn’s disease exclusion diet (CDED) [87,88]. The dietary intervention aimed at achieving remission consists of two phases, each of them lasting 6 weeks. The first phase is more restrictive and assumes that 50% of the caloric demand is provided from the formula. The second half of the demand is provided from the diet which consists of obligatory products which have to be eaten every day and complementary products proven to have no negative impact on the intestinal mucosa. Compulsory products include 150 g of chicken breast, two eggs, two bananas, one apple, and two steamed and cooled potatoes. In the second phase of the CDED, the compulsory products remain unchanged while the list of complementary products is extended and the caloric demand to be provided from the formula is reduced to 25%. Avoidance of products with proven proinflammatory effects continues to be the basic premise of the diet [87].

In a study by Sigall et al., the efficacy of a 6-week Crohn’s disease exclusion diet in achieving clinical remission of Crohn’s disease was at 70%, with a clinical response being obtained in 78% of subjects. In addition, remission was also achieved in six out of seven patients in whom the CDED was used as the only source of nutrition without the use of liquid formula [75]. The efficacy of the CDED has also been demonstrated in patients who had ceased to respond to biological treatment. Among 21 patients included in one study, dietary intervention led to remission in 62% of subjects and to a reduction in inflammatory markers in 81% of subjects [89].

A study comparing the efficacy of exclusive enteral nutrition with a combination of the CDED and partial enteral nutrition in a pediatric population revealed a marked superiority of the CDED with PEN in terms of tolerance and compliance with no statistical differences being observed in the efficacy of the dietary interventions of interest regarding clinical remission after 6 weeks of treatment. After 12 weeks, when the dietary restrictions were loosened for EEN patients, remission maintenance was significantly higher in the CDED/PEN group [22].

The efficacy of the Crohn’s disease exclusion diet was also demonstrated in a population of adult subjects by Szczubełek et al. [90]. After 6 weeks of dietary intervention, clinical remission was achieved in 76.7% of patients, with the percentage increasing to 82.1% after 12 weeks of treatment. This effectiveness was also confirmed by Yanai et al. in a randomized trial on an adult population of CD patients. This study showed that both the CDED plus PEN and CDED alone were effective in inducing and maintaining remission of CD [91].

The use of the CDED plus EEN, as well as EEN, resulted in changes in the composition of the gut microbiome in patients. These changes were greater among patients who achieved disease remission during the intervention. This suggests a significant impact of the modification of the intestinal microbiome, as a result of the applied nutritional intervention, on its effectiveness [22].

## 5. The Role of Diet in Functional Disorders during the Remission of Inflammatory Bowel Diseases

The natural history of IBD consists of periods of remission intertwined with periods of active disease. In some patients, gastrointestinal symptoms continue to be observed during disease remission. However, the symptoms are a manifestation of the gastrointestinal functional disease (GIFD) rather than of the primary disease. The most commonly reported GIFD is irritable bowel syndrome (IBS).

Irritable bowel syndrome is a heterogeneous, chronic condition with multifactorial etiology. The most important pathophysiological factors responsible for the development of IBS include quantitative and qualitative disturbances in the composition of gut microbiota, intestinal motility disorders, and visceral sensory disorders [92,93]. IBS is diagnosed on the basis of Rome IV criteria. The coincidence of certain gastrointestinal symptoms and their duration is taken into account; such symptoms include, e.g., recurring abdominal pain, constipation, or diarrhea [94].

The prevalence of IBS in the overall population varies from 10 to 25% depending on the diagnostic criteria. In patients in remission of Crohn’s disease, it is up to 36%, and in patients with quiescent ulcerative colitis 28% [95].

In the NutriNet-Sante Study [96] cohort of 33,000 patients, symptoms of IBS were shown to be dependent on diet, and the intensity of IBS symptoms was shown to increase with the increased intake of processed foods. 

The use of a gluten-free diet is popular among IBS and IBD patients. However, the rationale for its use and its efficacy outside celiac disease is controversial. Results of some studies on the efficacy of a gluten-free diet in IBS are indicative of some cereals rather than gluten being responsible for symptoms in patients. The symptoms are triggered by fermentable oligo-, di-, monosaccharides, and polyols (FODMAPs) which are found in large quantities in wheat, spelt, barley, and rye, i.e., cereals which also contain gluten [97,98].

The low FODMAP diet is effective in reducing the symptoms of irritable bowel syndrome. Studies have shown that the low FODMAP diet can reduce symptoms in up to 70% of IBS patients [99,100]. In our literature survey, we were unable to identify any studies suggesting the lack of efficacy of the low FODMAP diet in terms of reducing the IBS symptoms, with only the study by Bohn et al. pointing to the efficacy of the low FODMAP diet being comparable to that of standard diets used in IBS patients [101].

The low FODMAP diet is an elimination diet in which products rich in fermentable ingredients such as fructans in cereals, fructose in fruit, or lactose in dairy products are excluded from the patient’s menu. It should be noted that being an elimination diet, the low FODMAP diet should not be followed for periods longer than 6 weeks. Extending this period may lead to a worsening in gut dysbiosis or nutritional deficiencies [102,103].

The benefits of the low FODMAP diet in the remission of inflammatory bowel diseases were proven in a meta-analysis of nine studies. It reduces functional gastrointestinal symptoms (FGSs) but has no effect on stool consistency and fecal calprotectin (FC) [104].

Cox et al. [105], in a randomized study on the British population, proved the effectiveness of the low FODMAP diet in the reduction of intestinal symptoms among 52 patients in remission of IBD. Similarly, Bodini et al. [106] showed that the use of a periodic diet with the restriction of fermenting saccharides and polyols by patients with quiescent IBD is associated with improved quality of life and amelioration of inflammatory markers.

However, it should be remembered that the low FODMAP diet is associated with a reduction in a large number of vegetables, fruits, and grains, which may contribute to lower consumption of dietary fiber, and thus reduce the diversity of the intestinal microbiome and reduced SCFA production. The data available in the literature on this are unclear. Vandeputte et al. [107] describe a reduced abundance of Bifidobacetrium during the low FODMAP diet. Cox et al. [105] showed a lower number of Bifidobacterium longumi, Faecalibacterium prausnitzii, and Bifidobacterium adolescentis, but no differences in the diversity of the microbiome were found. On the other hand, Halmosa et al. [108] showed a reduced number of Akkermansia muciniphila and an increased number of Ruminococcus in the intestinal lumen, but also no differences in the number of bacteria [109].

The impact of the low FODMAP diet, as an elimination diet used to alleviate symptoms in IBD patients, requires further research into the risk of malnutrition, nutritional deficiencies, and changes in the composition of the gut microbiome.

Supplementation with microencapsulated butyric acid was also proven to be effective in the amelioration of mild IBS symptoms. The efficacy of sodium butyrate in the reduction of IBS symptoms was demonstrated by Lewandowski et al. [110]. The study was conducted on nearly 3000 patients, with statistically significant improvement being demonstrated in terms of the severity of abdominal pain, diarrhea, constipation, and bloating. In addition, reports of the beneficial effects of short-chain fatty acids on intestinal mucosa can be found in the literature, also in relation to inflammatory bowel diseases, as already mentioned before.

## 6. Other Elimination Diets in Inflammatory Bowel Diseases

Elimination diets such as specific carbohydrate diets [111,112] and the anti-inflammatory IBD-AID diet [113] are popular in the treatment of patients with inflammatory bowel disease. However, none of these diets has made it into the IBD management guidelines and therefore routine use of these diets in IBD patients is not recommended.

It is common in patients with IBD to eliminate lactose, a sugar derived from dairy products, from the diet. Crooks et al. [114] in 2021 showed that 21% of patients with ulcerative colitis in remission eliminated lactose from the diet. The elimination of lactose and dairy products, along with the elimination of gluten, is one of the most frequently used dietary restrictions among IBD patients [115]. However, it is noted in the literature that milk sugar intolerance assessed by breath test and analysis of genetic polymorphism among healthy controls is comparable to the group of IBD patients. Therefore, there is no need to routinely eliminate lactose from the patients’ diet, unless required by the patient’s specific dietary intervention, e.g., CDED [116].

New dietary interventions dedicated to patients with inflammatory bowel disease continue to appear in the literature. One of these is the ulcerative colitis exclusion diet (UCED) [117]. UCED is an elimination diet aimed at inducing remission in patients with ulcerative colitis. The diet is aimed at the modification of gut microbiota. For the time being, its effectiveness has been proven only in the pediatric population, with one study showing remission in 37.5% (9 out of 23) of patients. In addition, antibiotic therapy was pursued in eight patients in whom no remission had been achieved as the result of the UCED, resulting in remission being achieved in another four patients [117]. Notably, the UCED is a very strict elimination diet. Only vegetables, fruit, rice, potatoes, and certain quantities of chicken, yogurt, and eggs, are allowed during the first interventional phase of 6 weeks. Due to such a strict nature of dietary restrictions, the efficacy of the diet must be confirmed in a larger group of subjects, including adult subjects, and the impact of the diet on patients’ nourishment status must be assessed before the diet is broadly recommended for the management of patients.

Figure 1 summarizes the dietary interventions, mentioned in the text, that are used in IBD and the interventions that are being studied for usefulness. 

## 7. Conclusions and Future Direction

A growing number of studies and researchers are paying attention to the influence of diet on the risk and course of IBD and the possibility of nutritional treatment of patients. This topic seems to be becoming more and more popular due to the fact that treatment through nutritional therapy is usually cheaper and has fewer side effects.

There are nutritional interventions that are proven effective in inducing remission of IBD or that support other therapies. Such interventions include the CDED, EEN, or Mediterranean diet.

It is necessary to design and conduct randomized trials, rarely conducted in the field of nutritional therapy, to evaluate the effectiveness of currently used diets in IBD. It is also necessary to learn more about the effects of food ingredients on inflammation, the gut microbiome, and gut homeostasis. This knowledge will allow the development of new dietary interventions.

An interesting and poorly studied dietary intervention is the UCED, which requires further research in a larger group of patients and in the adult population.

Much recent research has focused on the effects of diet on the gut microbiome, which is a very complex “organ” in the human body. The function of the intestinal microbiome among IBD patients is not uniform, which is associated with the need to personalize the therapy used to change it, including nutritional therapy. The effectiveness of the applied therapy may depend on genetic and environmental factors. It seems that personalizing the approach to the selection of therapy, including nutritional therapy, is the direction in which the increasingly popular precision medicine should develop.

The development of biomarkers and prognostic tools will identify patients who will benefit from nutritional therapy. It is very likely that such a tool will be the analysis of the intestinal microbiome and the selection of interventions based on the desired changes in its composition in a particular patient.

## Figures and Tables

**Figure 1 nutrients-14-04965-f001:**
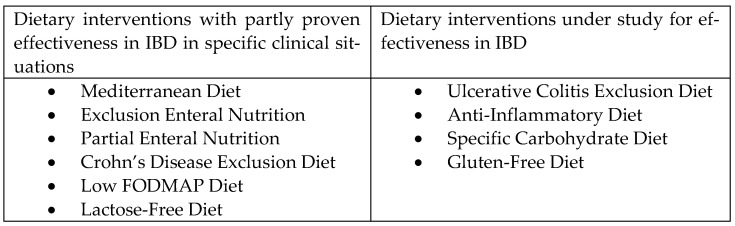
Dietary interventions included in the article that are applicable or potentially applicable in inflammatory bowel disease in specific clinical situations [21,22,43,72,73,74,75,76,77,84,85,86,87,88,89,90,91,99,100,101,104,106,111,112,113,116,117].

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
