# Peer review of "Selected Aspects of Nutrition in the Prevention and Treatment of Inflammatory Bowel Disease"

_nutrients, 2022, doi:10.3390/nu14234965_

Round 1

Reviewer 1 Report

Some of the comments about microbiota changes in both Crohn disease and ulcerative colitis could be more specific and it would be desirable to make a more comprehensive criticism on its influence by different modifications of the diet received.

It would be interesting to compare different results in children and adults with different diets and its effect in disbiosis described in the literature revisited.  

Author Response

Thank you for your time. We added comments about changes in the gut microbiome during dietary interventions. We expanded the section about gut microbiome.

We will be grateful for reading the article again.  

Reviewer 2 Report

Dear authors,

this review need some major revisions, as detailed below:

1.    Consider data about lactose free-diet

2.    Comments about prebiotics/probiotics maybe useful.

3.    Comments about  micronutrients supplementation should be added (see and cite  “Siva, S.; Rubin, D.T.; Gulotta, G.; Wroblewski, K.; Pekow, J. Zinc Deficiency is Associated with Poor Clinical Outcomes in Patients with Inflammatory Bowel Disease. Inflamm. Bowel Dis. 2017, 23, 152–157.”).

4.    I suggest to add a c omment about obesity and the need for nutriotional counseling in obese IBD patients: patients with IBD and obesity should be encouraged to lose body weight during the remission phase to improve the course of the disease, reduce obesity‐related comorbidities, and enhance response to therapy with biologicals. (see and cite “Bischoff SC, et al. European guideline on obesity care in patients with gastrointestinal and liver diseases - Joint European Society for Clinical Nutrition and Metabolism / United European Gastroenterology guideline. United European Gastroenterol J. 2022 Sep;10(7):663-720.”)

5.     Some recent interesting papers should be cited: Yanai H, et al. The Crohn's disease exclusion diet for induction and maintenance of remission in adults with mild-to-moderate Crohn's disease (CDED-AD): an open-label, pilot, randomised trial. Lancet Gastroenterol Hepatol. 2022 Jan;7(1):49-59; Adolph TE, Zhang J. Diet fuelling inflammatory bowel diseases: preclinical and clinical concepts. Gut. 2022 Sep 16:gutjnl-2021-326575;Yan J, et al. Dietary Patterns and Gut Microbiota Changes in Inflammatory Bowel Disease: Current Insights and Future Challenges. Nutrients. 2022 Sep 27;14(19):4003.

6.    A figure summarizing the types of diets analyzed in the text may be added.

Author Response

Thank you for your time.
I added a comment about obesity, lactose and probiotics. I have read and quoted the proposed literature.

I will be grateful for reading the article again.